# Recent Advances in the Detection of Antibiotic and Multi-Drug Resistant *Salmonella*: An Update

**DOI:** 10.3390/ijms22073499

**Published:** 2021-03-28

**Authors:** Siying Wu, John P. Hulme

**Affiliations:** 1Department of Biomedical Engineering, City University of Hong Kong, Hong Kong; siyingwu2-c@my.cityu.edu.hk; 2Department of Bionano Technology, Gachon Bionano Research Institute, Gachon University, Seongnam-si, Gyeonggi-do 461-701, Korea

**Keywords:** multi-drug resistant, *Salmonella*, detection, phenotypic, genotypic

## Abstract

Antibiotic and multi-drug resistant (MDR) *Salmonella* poses a significant threat to public health due to its ability to colonize animals (cold and warm-blooded) and contaminate freshwater supplies. Monitoring antibiotic resistant *Salmonella* is traditionally costly, involving the application of phenotypic and genotypic tests over several days. However, with the introduction of cheaper semi-automated devices in the last decade, strain detection and identification times have significantly fallen. This, in turn, has led to efficiently regulated food production systems and further reductions in food safety hazards. This review highlights current and emerging technologies used in the detection of antibiotic resistant and MDR *Salmonell*a.

## 1. Introduction

*Salmonella* is a Gram-negative bacterium accounting for 41% of diarrhea-associated deaths globally [1,2] The highly diverse pathogen is divided into two species: *Salmonella bongori* and *Salmonella enterica.* More than 2600 serotypes have been catalogued, with 1500 of those belonging to six subspecies of *S. enterica* (*Subsp. enterica*), namely *enterica* (I), *salamae* (II), *arizonae* (IIIa), *diarizonae* (IIIb), *houtenae* (IV), and *indica* (VI) [1]. Each serotype is identifiable via a unique variable region of the lipopolysaccharide (LPS) O-antigen and the flagellin structural proteins. Causative serotypes vary from continent to continent, with typhoidal Salmonellae (serotypes Typhi and Paratyphi A) common in South East Asia and non-typhoidal *Salmonella* (NTS) serovars common in Africa [3]. The infectious doses for typhoid and non-typhoid salmonellosis are 1000 colony forming units (CFU) and 1 CFU, respectively. NTS serovars generally induce mild gastroenteritis; exceptions include *S*. Dublin and *S*. Chloleraesius, which can result in bacteremia [4,5,6]. The severity of *Salmonella* disease (salmonellosis) depends on several factors, such as serotype (typhoid, non-typhoidal), gut colonization resistance, and the host’s immunosusceptibility to intracellular infection. Currently, there are four toxins produced from *S. enterica,* namely, SpvB, ArtAB, SboC/SeoC, and typhoid, all exhibiting ADP-ribosyltransferase activity resulting in actin depolymerization in the host [7,8,9,10].

According to recent reports by the Center for Disease Control and Prevention (CDC), the largest recorded *Salmonella* outbreak occurred in the USA in 2019, with 1134 reported cases, 219 hospitalizations resulting in the death of two people [11]. Serotypes Typhimurium, Newport, Heidelberg, and Hadar are currently listed by the CDC as “most threatening to public health” due to their frequent adulteration of beef and poultry food products and their association with multi-drug resistance [12,13]. The accepted definition of MDR is co-resistance to three or more classes of antimicrobial drugs [12]. Multiple molecular assays based on the detection of conserved genes *invA* and *ttrC* have been developed for the overall detection of the *Salmonella* genus [2]. Whereas genotypic identification of antibiotic resistance strains primarily focuses on the detection of integrons.

The most common integron found in isolates associated with multi drug resistant (MDR) is type I. Although antibiotic stewardships [14] have significantly curbed the number of *Salmonella* isolates exhibiting resistance to broad-spectrum antibiotics (Figure 1), isolates from US chicken and turkey foods frequently harbor plasmids for extended-spectrum β-lactamases (ESBLs).

Many of these plasmids also encode for *bla_CTX-M_* and *bla_SHV-5_* gene derivatives, conferring resistance to 3rd generation or last line cephalosporins. Some of the MDR genes present in various serotypes isolated from food products (poultry, swine, beef) and humans are shown in Appendix A [15].

The success of antibiotic stewardship programs in reducing the prevalence of MDR genes in foods is clearly evident (Figure 1). However, recent reports from India suggest [16] that antibiotic alternatives (dietary probiotic supplements) known to reduce MDR *S.* Typhimurium colonization in pigs can harbor resistance to many antibiotics, including ceftazidime, when misused. Moreover, a recent outbreak of Salmonellosis in China involving MDR *S.* Typhimurium [17] further highlighted the need for training and rigorous monitoring of these programs. Gene analysis of the serovar showed it contained several resistant genes, including *bla_OXA-1_*, *bla_TEM-1_,* and *β-lactamase*. In addition, an MDR *S*. Typhimurium co-harboring *mcr-1*, *fosA3*, *bla_CTX-M-14,_* was recently detected in the feces of a food production worker in China [18]. Thus, in order to better evaluate the roles of humans and animals in antibiotic resistance dissemination, the application of in-field rapid sero specific assays are urgently needed [19]. Recently, advances in nucleic acid automated amplification technologies, whole genome sequencing (WGS), phenotypic testing, and smartphone technologies have resulted in quicker identification times [20,21,22]. This paper aims to review the various genotypic and phenotypic techniques used to identify and monitor antibiotic and MDR resistant *Salmonella* and present emerging biosensors in the field.

## 2. Conventional Isolation, Enrichment and Detection Methods

The cultivation of potential antibiotic resistant *Salmonella* colonies from human fecal and food samples involves five stages: pre-enrichment incubation, enrichment, plating, screening, and confirmation [4]. Pre-enrichment incubation (24 h) employs non-selective (35 °C) media such as peptone water and lactose broth to expedite the recovery of sub-level injured *Salmonella*. Although for resistant *Salmonella* and cells that have entered the viable but non-culturable (VBNC) state, an antibiotic or selenite broth may be preferred. The pre-enriched media is then inoculated into selective media containing inhibitory bile salts such as thiosulphate malachite green, thiosulphate, sulphamethazine or novobiocin promoting the growth of *Salmonella* over other bacteria [23,24]. Alternatively, official (Bacteriological Analytical manual (BAM)) inhibitory media such as Rappaport–Vassiliadis (RV) or tetrathionate broth can be employed. After sufficient enrichment (10^4^ cells mL^−1^), cells from the chosen media are plated onto selective agars. Commonly used agars include xylose-lysine-deoxycholate agar (XLD), brilliant green agar (BGA), *Salmonella–Shigella* agar (SS), bismuth-sulfite agar (BSA) and Hektoen enteric (HE). Colonies from different serotypes present in various colors black (*S*. Typhi) or pink (*S*. Arizonae), depending on the agar medium. Occasionally, serotypes (*S*. Montevideo) are not distinctive and are even missed (Lac+ *Salmonella enterica* Virchow, *S. enterica* Newport *or S. enterica* Typhi) and cannot be reliably identified at this stage of the process. After plating, suspected *Salmonella* colonies undergo incubation in/on a group specific media such as a triple sugar iron (TSI) or lysine iron agar (LIA) slant [23] followed by a confirmatory test for urease negative cultures. Cultures giving typical *Salmonella* reactions are then selected for further biochemical and serological testing. However, the same serotype can vary with antigenicity. In such cases a polyvalent *Salmonella* O antisera or a polyvalent *Salmonella* bacteriophage O1-OE serological confirmatory test maybe also be employed [6,23,24]. It should be noted that antimicrobial resistant *Salmonella* spp. can be recovered by incorporating antibiotics in pre-enrichment and selective media [25].

### 2.1. Rapid Salmonella Detection Methods

#### 2.1.1. Enzyme-Linked Immuno-Sorbent Assay (ELISA)

ELISA is an optical-based immunoassay technique traditionally used in serotype identification. There are three assay formats direct, indirect, and sandwich, with the latter preferred due to its lower limit of detection (LOD). ELISA can be conducted on inexpensive paper or plastic, permitting its application in economically deprived regions [26]. In affluent areas of the world commercial assays such as the *Salmonella* ELISA Test SELECTA/OPTIMA™ (Bioline APS, Vejile, Denmark), Assurance GDS™ for *Salmonella* (BioControl Systems, Inc., Bellevue, WA, USA), TECRA *Salmonella*™ (Tecra International Pty Ltd., French Forest, New South Wales, Australia), and Vitek Immuno Diagnostic Assay System™ (VIDAS) (BioMerieus, Hazelwood, MO, USA) and *Salmonella* R-Biopharm™ dominate the field. However, the majority of these assays require optimized samples (minimization of protein, fat and microbial interferents), necessitating long enrichment steps (12–44 h) [27] Once optimized, the detection limit (10^4^ to 10^5^ CFU/mL or 1 CFU/25 g) for the pathogen is achievable in as little as 2 h. In addition to ELISA, there are simpler immunoassay formats based on affinity chromatography, diffusion, and latex agglutination, some of which can detect multiple *Salmonella* serotypes in a single run. For example, recent work using an inexpensive lateral flow sensor [28] showed that five serogroups (O:2, O:3, O:4, O:7, and O:9) could be detected in less than 15 min. In addition, commercial latex agglutination assays such as “color *Salmonella*” (Wellcolex, Merseyside, UK™) remain very popular due to their reduced cost, ease of use, and high (99%) sensitivity for *Salmonella*. However, most of the stated immunoassays are only applicable to specific food types (processed or raw) with raw food taking significantly longer to analyze. Exceptions include fluorescence ELISA assays (VIDAS SLM™ plus ICS and EIA Foss™) available from bioMerieux Vitek and Foss Electric, which incorporate an immuno-separation step rendering them applicable to all food products [6].

Currently, there are no Federal Drug Agency (FDA) approved immunoassays [29,30,31] that can simultaneously detect the presence of NTS toxins SboC/SeoC SpvB, ArtAB in foodstuff, blood, or feces. Thus, possible guidance on food supplements and infectious treatments (synbiotics and antibiotics) remains limited.

#### 2.1.2. Nucleic Acid Assay Techniques

The most common nucleic acid technique used to identify *Salmonella* serotypes is polymerase chain reaction (PCR). Several validated and standardize (International Organization for Standardization (ISO)) PCR methods are already in use by the food industry (ISO 22174, OS 20838:2006, ISO 16140:2003, ISO/DIS 22119, ISO 6579:2002,) [32]. There are many PCR techniques such as real-time PCR (RT-PCR) and droplet (dd) PCR, which can provide results in less than 24 h in a laboratory setting. In addition, there are a variety of commercial instruments (Bio-Rad QX100 droplet digital PCR system™) that can identify and reliably quantify *Salmonella* in food, cells, and water. For example, in the detection [33] of the *ttr* gene, ddPCR recently exhibited a sensitivity of 2 GC/PCR in purified water compared to 20 CFU/PCR for RT-PCR. More recently, a two-step real time PCR process was developed in which the four most threatening serotypes [34] to public health were detected at low pre-enrichment concentrations (0.156 CFU/g of ground beef) within 4 h. In addition, the authors concluded that the procedure could be performed without an isolation step, further reducing the processing time. In 2017 the advantages of automated RT-PCR were again utilized in the rapid screening of 154 suspected MDR *Salmonella* isolates sampled from large-scale chicken, duck, and pig farms [35]. Out of the 154 isolates, 26 possessed the class 1 integron containing gene cassettes *drfA17- aadA5, drfA12*-*aadA2*, *aadA2*, *aadA1*, and *drfA1-aadA1*, while more than a third of the isolates (55/154) also carried *bla_TEM−1_*.

Multiplex PCR (MPCR) is a technique that permits the amplification of numerous DNA targets making it well suited for detecting antibiotic resistant *Salmonella* in serovar rich foods. As such, several groups have employed triplex [36,37] or even (Taq-Man^®^real-time PCR) pentaplex M-PCR [38] in the simultaneous detection of virulent (invasin virulence (invA)) and antibiotic resistant genes. Commercial platforms (Taq-Man^®^real-time PCR) used in the detection of MDR *Salmonella* routinely exhibit an LOD of 10 CFU/g for enriched samples isolated from beef trim, tomato, eggs, and spinach. Although the use of multiple targets is advantageous, MPCR remains sensitive to proteinous interferents that delay the Cq (threshold cycle), causing erroneously low estimates of the template. In addition, cross-reactivity of primer pairs can arise during amplification, imposing further limitations on sensitivity. Currently an in-field MDR platform that can detect all four CDC listed serovars, as well as tetracycline and streptomycin resistance has not been reported.

Automated RT-PCR and digital PCR assays are expensive and are not applicable outside food emergency response laboratories. A cheaper alternative to the stated techniques and better suited to in-field testing is loop-mediated isothermal amplification (LAMP). LAMP is a simple amplification technique that relies on the auto-cycling strand dis-placement of DNA synthesis performed by Bst DNA polymerase and a set of primers that fold and create dumbbell DNA structures that trigger cycling isothermal amplification. Due to its ease of use and compatibility with inexpensive substrates, LAMP has emerged as a viable alternative to PCR, particularly in low resource settings [39]. Another advantage of LAMP is its tolerance for complex liquid or solid media, which also translates into food testing for pathogens. Figure 2 shows a pocket-sized paper LAMP device employed in the detection of *S*. Typhimurium.

In addition to labeled assays, non-labeled acoustic and optical LAMP assays have been used in the rapid detection (<1 h) of *S*. Typhimurium [41,42] in blood, saliva, nasal fluid (4 × 10^3^ CFU/mL) and urine (5 CFU/mL) as well. In addition to traditional LAMP, multiplex LAMP assays are just beginning to be explored [22] For readers seeking an in-depth perspective on the commercial development of LAMP and its current diagnostic applications in food and feeds the following reference is recommended [22].

## 3. Automated Whole Genome Sequencing

Whole genome sequencing (WGS) is a powerful, inexpensive open access epidemiological tool that can predict the genotypic and phenotypic resistance of a suspected bacterium in just a few days. Multi-locus sequence typing (MLST), multiple-locus variable number of tandem repeats analysis (MLVA), single-nucleotide polymorphism (SNP) analysis, [43], CRISPR-multi-locus virulence sequence typing (CRISPR-MVLST) [44] and next-generation sequencing (NGS) are some of the techniques [45] used in the sequencing of NTS antibiotic-resistant gene clusters, see Figure 3.

Upon acquisition, sequences are then compared to reference sets sourced from databases such as (ResFinder [Center for Genomic Epidemiology, DTU]), BLAST (blastn) and ARG-ANNOT, followed by a phenotypic test to validate the predicted accuracy. Typically, an isolate will be genotypically resistant if a suspected antimicrobial resistant (AMR) gene is 75% identical to a reference sequence. Mismatches naturally arise, resulting in errors; Broadly, there are two mismatch categories, very major errors (VME) and major errors (ME). VME’s occur when a microbial sequence is predicted to be genotypically susceptible yet is phenotypically resistant. Conversely, major errors arise when a microbe identified as genetically resistant expresses phenotypic susceptibility [46].

ME discrepancies seem to be associated with the breakpoints used for phenotypic testing. In some cases, the minimum inhibitory concentrations (MIC) are just below the recommended breakpoints suggesting technical variations in the agar dilution method may result in isolates being falsely classified as susceptible. Moreover, some plasmids encoding for antimicrobial resistance genes can be damaged during storage and sub-culturing, further affecting the ME rate. In the VME category, mismatches are generally attributed to the presence of resistance determinants absent in the reference database or novel resistance mechanisms whose genetic determinants have yet to be determined.

These mismatches seem to be a relevant issue [47], especially when predicting streptomycin resistance. Despite these issues, ME and VME rates from numerous antibiotic resistant *Salmonella* WGS studies remain below the acceptable FDA cut-offs of 3 and 1.5%. In an extensive WGS study [48] involving 640 NTS Salmonellae, the susceptibility of 43 different serotypes to 14 antimicrobials was tested. Using the phenotypic results as the reference outcome, the authors calculated sensitivities by dividing the number of isolates that were genotypically resistant by the total number of isolates exhibiting clinical resistance phenotypes. Specificity was calculated by dividing the number of isolates that were genotypically susceptible by the total number of isolates with susceptible phenotypes. A total of 65 unique resistance genes plus mutations in two structural resistance loci were identified. Minimal sensitivity and specificity values of 86.4 (sulfisoxazole) and 90.8% (streptomycin) were reported.

In another WGS study, AMR *Salmonella enterica* serovars (Typhimurium, Newport, and Dublin) from 90 isolates sourced from humans and cattle were compared. Isolates were screened for phenotypic resistance to 12 antibiotics. Genotypic prediction of phenotypic resistance resulted in a mean sensitivity of 97.2 and specificity of 85.2, respectively [49]. Additional work [46] confirmed [48] previous observations regarding discrepancies between phenotypic resistance and genotypic resistance of aminoglycoside resistant genes. The authors concluded that 35 isolates carrying streptomycin resistance genes were phenotypically susceptible to the drug.

In 2019, a WGS study [50] involving multiple phenotypic susceptibility testing methods produced ME and VME rates of 4.5% and 17%. Fifty-one of the VME’s were the result of discordant sulfisoxazole and sulfamethoxazole predictions. Interpretation of seven antimicrobials using *Salmonella* clinical breakpoints produced low genotyping sensitivity and specificities values of 0.84 and 0.88, respectively. Upon exclusion of the VME rates for streptomycin and sulfamethoxazole, said values increased to 0.89 and 0.97. Notably, the WGS-based genotyping methods used in the study did not account for attenuation mechanisms or reliably predict for underlying temporary genetic features (tandem repeats) present in subpopulations (heteroresistance) during phenotypic testing [51].

Whether heteroresistance will prevent WGS from becoming the primary diagnostic tool for antibiotic resistant pathogens is debatable, given the techniques improving resolution and increasing usage by food and drug industries. It is worth mentioning that for the past thirty years, the evolution of MDR *S.* enterica serotype Kentucky and the subsequent emergence of the MDR clone in Africa and the EU were successfully mapped using WGS [52]. During that time, WGS has undoubtedly improved our understanding of serovar resistome profiles and the employment of smarter antibiotic resistance combat strategies.

## 4. Automated Phenotypic Testing

### 4.1. Manual and Semi-Automated Antimicrobial Susceptibility Test (AST)

AST is used to determine the MIC of an antimicrobial required to limit a pathogen’s growth in accordance with guidelines provided by the Clinical and Laboratory Standards Institute (CLSI) or the European Committee on Antimicrobial Susceptibility Testing (EU- CAST in optimized media for a standardized period [53]. Traditionally that media is a broth or agar in which microdilution tests or disk diffusion tests between 35−37 °C are conducted. Tests performed using liquid-based methods measure change in optical density whilst disk diffusion methods estimate the antibiotic inhibition zone on agar plates following 24 h incubation. The size of the zone is a direct measure of the susceptibility of the bacteria to an antibiotic and is inversely related to the minimum inhibitory concentration MIC). Agar and broth dilution are some of the earliest techniques used to measure MIC. The advantages of broth dilution are its reproducibility and cost-effectiveness. A modern version of the technique called microbroth dilution, is now commonly employed.

Minimum inhibitory values are also determined by several other methods such as paper diffusion, E-tests, Biolog ^MT^ plate, turbidity and absorbance assays. Although relatively inexpensive many of these growth dependent methods are hindered by long incubation periods sometimes taking more than 16 h to complete [54]. As a matter of urgency, the FDA recently approved five semi-automated systems for clinical use, including Phoenix™ (Becton Dickinson Diagnostic Systems, https://www.bd.com/en-us, accessed on 1 February 2021), Sensititer ARIS 2X™ (Trek Diagnostic Systems, https://www.thermofisher.com, accessed on 1 February 2021), WalkAway™ (Siemens Medical Solutions Diagnostics, https://www.beckmancoulter.com, accessed on 1 February 2021), and the VITEK systems 2™, https://www.biomerieux-usa.com/vitek-2, accessed on 1 February 2021). The VITEK system employs three sensing modalities, absorbance, fluorescence, and turbidity, to determine an antibiotic’s MIC. The average time of an MIC for an antibiotic using the said system is 8 h. Recent work [55] showed that the performance (sensitivity and range) of the VITEK system could be significantly enhanced when a mono-sulfonated tetrazolium salt such as “EZMTT” is added to the growth medium. Alternatively, one can use the highly automated fluorescent-based DxM Micro Scan WalkAway™ system. The advantage of DxM is that each MicroScan Dried Gram-Negative (MSDGN) MIC panel is pre-packaged with a tailored growth medium, fluorogenic panel and an antibiotic. Hydrolysis of the fluorogenic panel correlates directly to enzymatic activity, permitting the inclusion of persistent *Salmonella* in the MIC measurement [56].

Isolation, identification, and AST procedures can take from 2–7 days, depending on the number of samples and availability of automated culturing equipment (ACE). For example, the “Walk Away” specimen processor unit (WASP™) and BDs Kiestra TLA™ or BACTEC MGIT™ can reliably detect antibiotic resistant *Salmonella* in a much shorter period compared to manual approaches [57]. In 2018 initial trials of the Accelerated PhenoTest BC™ (Accelerated Diagnostics) showed it was possible to combine the identification and AST stages in a single test [58].

The PhenoTest utilizes a combination of fluorescence in situ hybridization (FISH) and electro-kinetic focusing to concentrate and identify bacteria. The concentrated cells then undergo analysis via automated dark field microscopy (ADFM). ADFM utilizes a series of stacked z-images to monitor changes in colony shape, surface area, and segmentation. The compiled image permits the differentiation of bacterial growth, death, and elongation. When combined with deep learning processes, the Accelerated PhenoTest BC™ and the recently introduced QMAC-dRAST™ (QuantaMatrix, Inc.; Seoul, Korea) can generate a MIC in less than 7 h. Unfortunately, the FDA recalled the Accelerated PhenoTest BC™ kit in late 2018 due to the high number of false positives [58]. For readers seeking an in-depth perspective regarding commercial AST testing, multiple reviews are recommended [54,58].

### 4.2. Detection of Intracellular Resistance Salmonella Using Flow Cytometry

Flow cytometry (FC) is an optical technique that measures the fluorescent and scattering properties of a laser integrated single line of cells as they continuously flow by a detector or detectors [59]. Differing flow rates can be used to study antibiotic-induced changes in cellular morphology, intracell heterogeneity, cell-to-cell interaction (e.g., quorum sensing), and in the analysis of subpopulations of persistent and moderately resistant strains of *Salmonella* [60]. As well as persistence, FC has also been used to measure the transfer ratios of the multi-antibiotic resistant plasmid pB10 in multiple strains of *Salmonella* [61] and reliably determine antibiotic susceptible gram-negative phenotypes albeit in the absence of mammalian cells [62]. In addition to susceptibility, the re-potentiation of antibiotics in the presence of primary metabolites and subsequent elimination of persistent *Salmonella* from macrophages was also evaluated [63] with FC.

Elimination of persistent *Salmonella* residing in the lymph nodes of animals and humans remains a costly and significant challenge. Some of the initial work in 2014 [64] using FC, a mouse typhoid fever model and the single-cell growth reporter (DsRed S197T) showed that differential host nutrient supply contributed to the heterogeneity of *Salmonella* subsets. Assessment of the antimicrobial tolerance of the various subsets showed overall eradication was delayed primarily by abundant moderately growing *Salmonella* with partial tolerance.

In another study, GFP-expressing *S*. Dublin (SD3246-GFP) was used to estimate the degree of intracellular infection in bovine-derived macrophages sourced from 3 and 28-day-old Friesian bull calves. Confirmation by gentamicin-protection assay showed *S*. Dublin intracellular replication and survival were arrested after 6 h and up to 24 h, respectively [65]. Interestingly, FC results revealed that the majority of infected cells expressed MHCII, CD40, CD80, CD86, CD11b and CD11c but did not express CD1 Further analysis revealed that infected MHCII+ macrophage-like cells expressed elevated levels of MHCII and CD40 compared to uninfected cells, which is in contrast to observations reported with murine models. The authors attributed these inconsistencies to unnatural target species used in previous studies [66].

Given FC’s ability to simultaneously measure susceptible, intermediate, and resistant (SIR) phenotypes, several initiatives have been proposed by various national, European, and international bodies to promote the development of rapid FC based AST assays. Of note is the commercialized FASTvet assay (FASTinov^®^) developed by the Fast-Bac European consortium. In a recent study, the kit-assay exceeded expectations demonstrating an AST turn-around time of 2 h for 13 antimicrobials [67], which is significantly faster than dRAST and many other automated phenotypic and genotypic platforms described herein. However, the authors did note that further testing using whole blood samples was required for veterinary usage.

Flow cytometry remains a highly versatile measurement tool, capable of determining the susceptibility of pre-enriched bacterial isolates, the transfer ratios of antibiotic resistant plasmids in multiple strains of *Salmonella* and mammalian cells that harbor antibiotic resistant species. Moreover, with the advent of the FASTvet assay, FC has the potential to rapidly identify and monitor antibiotic resistance bacteria in food animals.

## 5. Emerging Biosensors

Depending on the application (microbiology or biological), many definitions of a biosensor have been reported in the literature [68,69]. According to the International Union of Pure and Applied Chemistry, a biosensor is an integrated receptor (protein or DNA) transducer device capable of providing specific quantitative or qualitative information via a biological recognition element (BRE). Simply put, when a specific analyte binds to a receptor, a transfer (mass, electrical, thermal, magnetic, or photonic energy) occurs between the two, which is detected via a prescribed transduction element (e.g., electrode pattern, optical waveguide, fiber, cantilever, etc.). The transduction element then relays the information to an integrated (small screen) or separate (human, laptop) monitoring component. A “device” is a single entity upon which binding, transfer, and monitoring occur. The fundamental parts of a biosensor are shown in Figure 4. Biological recognition elements used in microbial sensing include antibodies, toxins, nucleic acids, whole cells, and biomimetic materials [70,71,72].

### 5.1. Optical Biosensors

A variety of biosensors based on fluorescence, absorption, refractive index (RI). Raman, and Surface Enhanced Raman Spectroscopy (SERS) techniques have been used in the identification of antibiotic resistant *Salmonella* [73,74,75] over the last two decades. However, many unmodified nanomaterials known to enhance device sensitivity, such as nanoAg and carbon nanotubes (CNT), are antibacterial, potentially leading to false positive results when evaluating antibiotic sensitivities [76]. Thus, the use of nanoAg and the prescribed growth medium for the pathogen (including wild types) have to be carefully controlled. Moreover, the impact of reactive oxygen species (ROS) on downstream bacterial DNA amplification is still unknown, as is CNT’s suspected mutagenic (chromosomal) role.

These issues have caused some groups to revisit traditional molecular labels such as heavy water (D_2_O) and other radioisotopes in order to reliably discriminate between live and dead bacteria at the single-cell level [77]. The researchers conducted additional single-cell AST tests on several WHO pathogens, including *S. enterica*. Moreover, recent findings involving another radioisotope ^15^N showed it can be used to monitor the red-shift in the resonance Raman spectra of cytochrome C, a key player in many resistance pathways [78,79].

During the past twenty years ATP Assays particularly ATP Bioluminescence assays have become established in the meat industry due to their reliability (microbial counts) speed (15–20 min assay time) and ease of use [80,81]. Intracellular ATP in a sample gives an indirect measurement of the number of viable cells in a given sample also indicating whether those cells are stressed. Highly metabolically stressed bacteria have an intracellular ATP 10–30% of their optimized counterparts. In addition to intracellular ATP, extra cellular ATP (eATP) can be used to estimate viable cells in real time. In 2021 patterns of eATP determined by real-time luminescence measurement were used to infer the MIC of Ampicillin for (*Enterococcus faecium, Staphylococcus aureus, Klebsiella pneumoniae, Acinetobacter baumannii, Pseudomonas aeruginosa, and Enterobacter species*) ESKAPE pathogens [82]. In addition, recent studies have shown eATP to be a reliable indicator of bacterial susceptibility [83]. An alternative approach to eATP measurement would be to monitor the proton flux in the localized external environment of *S*. Typhimurium at the single cell level. This would be achievable via a single cell microdroplet smart gel assay, although such a highly automated digital approach would not be economically feasible outside the research laboratory. A transduction element that is sensitive to proton flux and may have in-field application is an F_0_F_1_-ATPase rotor. An investigation using an aptamer-based F_0_F_1_-ATPase biosensor consisting of a “rotator” e-subunit of F_0_F_1_-ATPase combined with an anti-e-subunit antibody-biotin-avidin-biotin-aptamer linker was used in the detection of *S*. Typhimurium. The sensitivity of the device was dependent on proton flux driven by F_0_F_1_-ATPase-mediated ATP-synthesis. The authors demonstrated a clinically relevant range of 10^1^–10^4^ CFU/mL and LOD OF 10CFU/mL for *S.* Typhimurium [84].

As previously stated, there are four serovars with high potential for MDR posing the greatest risk to animal and public health [34]. However, during the last decade researchers have primarily focused on *S*. Typhimurium due to its prevalence within the food industry and its ability to act as persistent reservoir for resistant genes. With the advent of refined nanoparticle signal technologies, it is now possible to detect *S*. Typhimurium at very low levels following enrichment. Just recently, guanidine-functionalized up-conversion fluorescent nanoparticles (UCNPs@GDN), tannic acid, and hydrogen peroxide (HP) were used to quantify seven pathogenic bacteria, including *S.* Typhimurium. Tannic acid and HP demonstrated synergistic behavior, significantly increasing the sensitivity of a fluorescent-based detection system. The authors noted tailored specificity of the UCNPs@GDN’s was readily achievable via the conjugation of a serovar-specific aptamer to its surface. The sensor showed a linear range of 10^3^ to 10^8^ CFU/mL and an LOD of 1.30 × 10^2^ CFU/mL for uniform mixtures of bacteria [85]. This approach has significant potential in MIC and AST testing as it uses ROS species, a known indicator of growth inhibition, and antibiotic resistance. Thus, many traditional ROS-based assays can readily be repurposed for AST. A recent example involved the detection of *S*. Enteritidis via a standard HRP assay, streptavidin magnetic beads, biotin-labeled antibody, and a nanoporous “microflower”. The developed assay could detect *S.* Enteritidis in tap water, milk, and cheese, and relay detection limits of 1.0 CFU/mL via a mobile phone in less than an hour [86].

With the advent of cheaper, highly conductive 2D dimensional nanomaterials, such as graphene and molybdenum disulfide, interest in the sensing applications of mature nanomaterials like CNT and nanoAg has begun to wane. Recent work [87] employing a 2D–0D heterostructure-based SERS platform in the rapid detection of *Salmonella* DT104 utilized an alternative transduction element consisting of a (2D) WS2 transition metal dichalcogenide (TMD) and a (0D) plasmonic gold nanoparticulate (GNPs) lattice. The authors showed the device could rapidly discriminate multidrug resistant *Salmonella* DT104 from *S.* Typhi using AST and Augmentin antibiotics up to concentrations of 100 CFU/mL. Figure 5 shows the change in Raman spectra of the different strains before and after exposure to Augmentin. DT104 resistant AMP Raman bands at 570 cm^−1^, 968 cm^−1,^ and 1386 cm^−1^ were absent in antibiotic sensitive *S.* Typhi samples. Figure 5b,c demonstrate the inverse relationship between ATP release and cell viability. However, perhaps most interesting is Figure 5d and the consistent rupturing of the poles of long and short *S*. Typhi. Unfortunately, the authors did not supply an SEM of ruptured DT 104 for comparison.

In the last two decades, many types of label-free refractive index techniques have been used to detect *Salmonella* [88]. However, by far, the most common is surface plasmon resonance (SPR). There are many portable versions (e.g., multi-fiber electrode) of SPR, which can simultaneously detect multiple serovars in the field. In 2018, a Ω-Shaped Fiber-Optic Probe-Based Localized Surface Plasmon Resonance (FOLSPR) Biosensor was used in the quantification of *S.* Typhimurium. The sensor detected *S.* Typhimurium down to 128 CFU/mL within a linear range from 5 × 10^2^ to 1 × 10^8^ CFU/mL [89].

It’s well established that the [90] crystallinity of bacterial outer membranes plays a critical role in antibiotic susceptibility and membrane resistance. Thus, biomimetic liquid crystal sensors are ideal for measuring the antibiotic susceptibility of *Salmonella* and other pathogenic species at various temperatures. Consequently, many groups are developing such devices for the detection of food pathogens [91], including *Salmonella*. This impetus has already led to the fruition of several commercial liquid crystal sensors, such as the Crystal Diagnostics Xpress System^®^ (CDx) (Crystal Diagnostics Ltd. Broomfield, CO, USA), which has recently received multiple AOAC accreditations from the FDA.

### 5.2. Electrochemical Biosensors

Modern electrochemical sensors are small, sensitive, robust, and mass-producible, making them a cost-effective option for persons in low resource settings. A recent publication [92] highlighted the current state of *Salmonella* electrochemical sensors within the food industry. As inferred by the authors, the continued success of “electrochemical sensors” stems from an endless supply of quality screen-printed electrodes (SPE) and biocompatible conducting nanomaterials, making them a cheap and sensitive option for tailored AST applications [93]. Other advantages of electrochemical devices include their ability to probe optically opaque samples [94], compatibility with AST testing, and more recently rapid ASTs (RASTs). RAST platforms can provide a result in a couple of hours and can be fabricated (screen printing) on paper substrates. Just recently, researchers fabricated a RAST platform by depositing resazurin crystals, pyrolytic graphite sheets and a Nafion membrane on paper substrates. By monitoring the differential voltametric signals, the MIC of ampicillin and kanamycin for K-12 was measured in as little as 60 min [95]. In 2020 another paper- based platform provided a confirmatory AST result in 5 h [96] which is significantly better than automated assays discussed herein.

Other groups continue to use this inexpensive, highly sensitive approach in the detection of *Salmonella*. Recently, L-lysine functionalized cerium oxide nanoparticle coated indium tin oxide (L-CeONP/ITO) electrodes were used instead of solid nanoAg to predict the susceptibility of gram-negative bacteria in less than 15 min via cyclic voltammetry with minimal ROS interference [97]. Changes in the anodic peak current response of optimized bacteria before and after the treatment of antibiotics were considered as antibiotic susceptible by the authors. The time of interaction between the bacterium and antibiotic (fixed concentration of 2 μg/μL) varied from 0 to 18 h. In 2018, aptamer specific 2D molybdenum di-selenide (MoSe2) fluorine-doped tin oxide electrodes were used in the detection of *S*. Paratyphi via cyclic voltammetry (CV) and differential pulse voltammetry (DPV). Electrochemical studies revealed an excellent linear detection range of 10^2^–10^10^ CFU/mL with a LOD of 1 × 10 CFU/mL and an R2 of 0.98, respectively [98].

There are many electrochemical techniques such as dielectrophoresis, electrochemical impedance spectroscopy (EIS), and DPV, which are well suited for measuring the antibiotic susceptibility of *Salmonella* and other pathogenic species. In 2015 a DPV resazurin subtractive inhibition assay was used to evaluate the antibiotic susceptibility of *E. coli* [99]. Following incubation, the authors achieved AST values in an hour. More recently, screen-printed carbon electrodes and resazurin were used to detect various concentrations of target bacteria including *Salmonella gallinarum*. The MIC for *S*. *gallinarium* was similar to that obtained with conventional AST testing [100]. Resazurin remains a highly versatile redox agent shown to be compatible with numerous electrochemical and optical inexpensive sensing platforms [101].

Finally, electrochemical immunodetection of either *Salmonella* or antibiotic (tetracycline, ceftiofur) residues in foods and liquid media using non-faradaic electrochemical impedance spectroscopy (EIS) are well known [102,103]. However, to the best of our knowledge, no reports regarding residual antibiotic subtractive inhibition assays and *Salmonella* using electrochemical or optical (SPR) techniques have been published. In such circumstances, the antibiotic binding signal would be inversely related to the degree of *Salmonella* resistance. A summary of the biosensors used in the detection of antibiotic resistant serotypes discussed herein is shown Table 1.

## 6. Conclusions

Throughout this review, the advantages and limitations of commercial and noncommercial genotypic and phenotypic technologies employed in the detection and monitoring of antibiotic resistant *Salmonella* and to a lesser degree MDR *S.* Typhimurium have been highlighted. Automated technologies naturally dominate the field, providing the researcher with the tools to accurately predict the phenotypic characteristics of 1000 s of genotypic compositions. Yet a gold standard remains out of reach due to discrepancies. A major source of these discrepancies is hetero-resistant *Salmonella*, which cannot be predicted for using current WGS software [51]. Thus, no primary diagnostic tool (including WGS) currently exists for MDR *Salmonella*, although recent improvements in automated pre-commercialized SIR technologies (FASTinov^®^) could lead to better predictive accuracies in the future [67].

WGS remains the primary surveillance tool for antibiotic resistant *Salmonella*; however, operational limitations (absence of skilled labor) prevent its use in low resource (agricultural) settings where the need is felt the strongest. Efforts to increase the applicability (via training) of WGS and NGS systems combined with stewardship programs and feed alternatives, (synbiotics) will inevitably result in further reductions in antibiotic resistance.

In contrast to automated technologies, the advantages of qualitative one-shot assays and emerging sensing devices for rapid infield serotyping and AST purposes are also being developed by several companies. Some of these devices have recently received FDA approval for use in the food industry (Crystal Diagnostics Xpress System^®^ (CDx)). However, many of the biosensors herein have not gone through regulatory approval. In addition, no FDA approved hand-held devices that can identify prevalent MDR serotypes (*S.* Typhimurium DT104) in foods have been reported.

In the advent, probiotics fail to significantly reduce AR *Salmonella*; combinatorial approaches involving vaccines, synbiotics, and phage treatments are actively being explored [104,105] as long-term interventions in production agriculture. The success of these approaches requires collaborations between the FDA and WHO to safely identify, scale-up, and apply new interventions as soon as possible, particularly as the global food industry enters the post-antibiotic era.

## Figures and Tables

**Figure 1 ijms-22-03499-f001:**
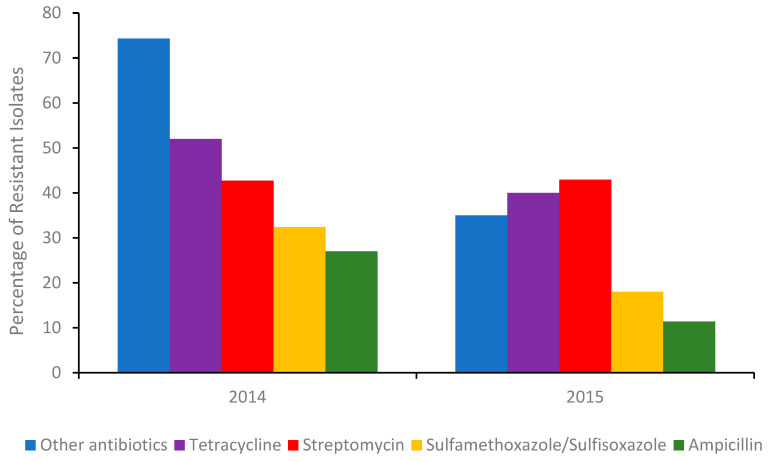
Percentage of the total number of *Salmonella* isolates in 2014 (Δ number of isolates 262) and the first half of 2015 (Δ number of isolates 114) detected in poultry, cattle, and swine meats containing a gene/genes (A table of those genes can be found in the Appendix A) conferring resistance to an antibiotic or other antibiotics. Sourced with permission (15). The percentage of isolates resistant to other antibiotics, were calculated from the sum of isolates conferring resistance to gentamicin, amoxicillin-clavulanic acid, cefoxitin, ceftiofur, ceftriaxone, azithromycin, chloramphenicol, ciprofloxacin, nalidixic acid, and trimethoprim-sulfamethoxazole respectively.

**Figure 2 ijms-22-03499-f002:**
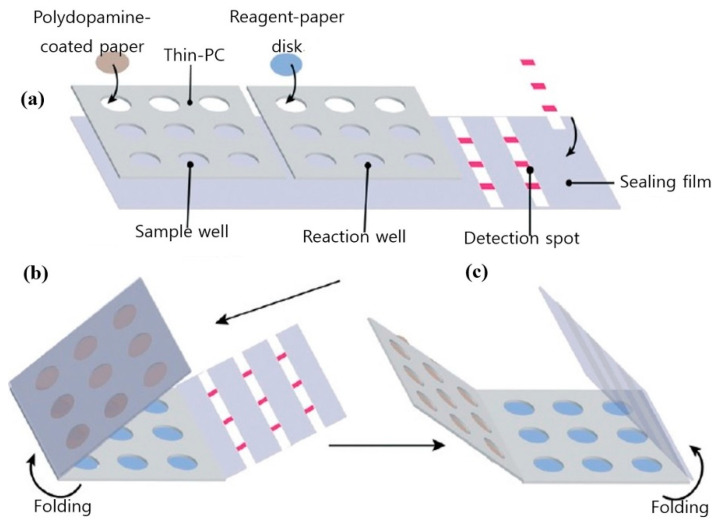
Design and fabrication of the foldable microdevice. (**a**) The microdevice’s three main components include a sample zone with nine sample chambers, a reaction zone with nine reaction chambers and a detection zone with three paper strips; (**b**) folding step for extraction; and (**c**) folding step for detection. Figure modified with permission from [40]. (Copyright 2019, Royal Society of Chemistry).

**Figure 3 ijms-22-03499-f003:**
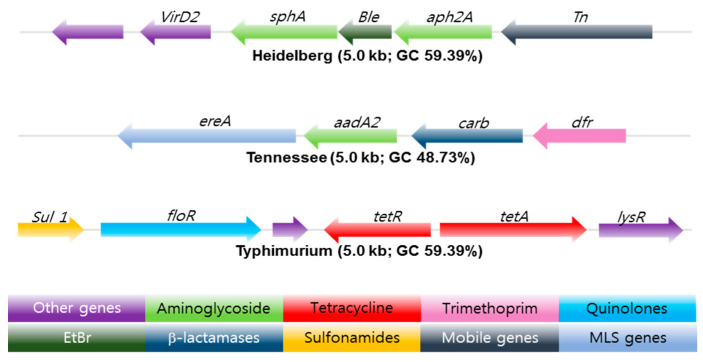
Antibiotic resistance gene clusters of three common *Salmonella* serotypes found in swine isolates from 2004–2005. EtBr resistance gene (Dark Green) flanked by antibiotic resistance genes marked in light green. The arrow shows the orientation of the genes; genes are color-coded to define different classes of antibiotic resistance, mobile, and other gene categories. Copyright permission PLOS One [45].

**Figure 4 ijms-22-03499-f004:**
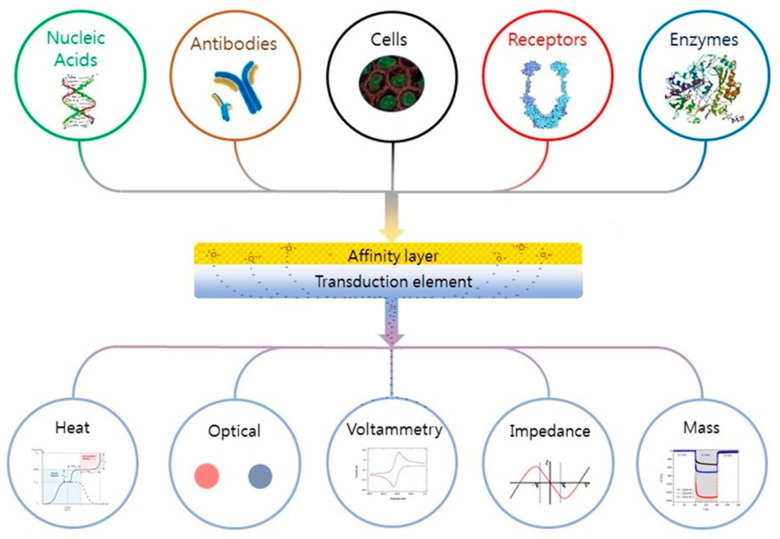
Schematic of a biosensor. Reading from the top down; A tailored affinity layer specific to an analyte or analytes. Binding of an analyte to the layer leads to changes in its physical properties (electrochemical, optical, heat, etc.), which are converted into a measurable signal by the transduction element; these are then relayed to and recorded by an electronic meter or visually (coloration) identified by the subject. Figure reproduced with permission from Elsevier [68].

**Figure 5 ijms-22-03499-f005:**
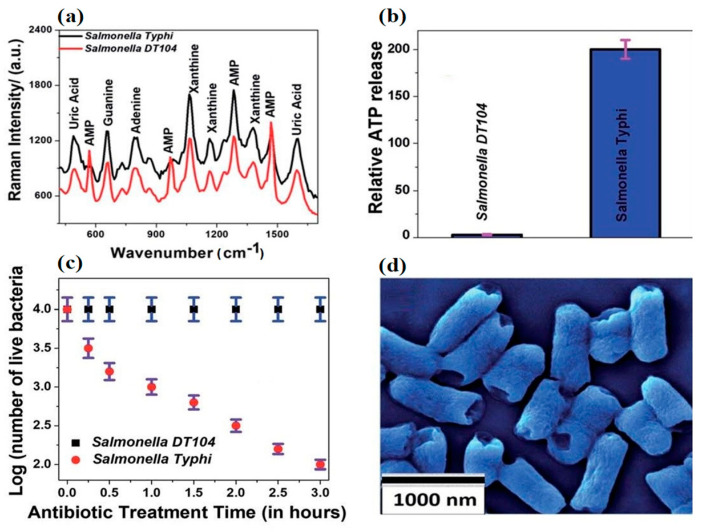
The Raman spectra of MDR *Salmonella* DT104 and a susceptible strain of *Salmonella* Typhi (**a**) on the heterostructure surface. Plots indicating the percentages of live bacteria for *Salmonella* DT104 and *Salmonella* Typhi after treatment with the (**b**) Augmentin antibiotic at 1000 CFU/mL. Plots showing the relative cellular ATP leakage from *Salmonella* DT104 and *Salmonella* Typhi after treatment with the Augmentin (**c**) antibiotic. An SEM image showing the damaged walls of *Salmonella* Typhi (**d**) after treatment with the Augmentin antibiotic. Figure modified with permission from [87]. (Copyright 2020, American Chemical Society).

**Table 1 ijms-22-03499-t001:** Biosensors used in the detection of antibiotic-resistant *Salmonella* serotypes. SERS (surface enhanced Raman spectroscopy).

*Salmonella* Serotypes	Sensing Method	Sample Matrix	Analysis Time (min)	Detection Limit (CFU/mL)	Reference
*S.* Typhimurium DT104	SERS	Assay media	30	10^5^	[71]
*S.* Typhimurium DT104	SERS	Assay media	15	10^5^	[74]
*S.* Typhimurium	SERS	H_2_O & milk	120	20	[72]
*S.* enterica	Raman Spectroscopy	Urine	150	n/a	[77]
*S.* Enteritidis,	Fluorescence	Water, milk, and beef	30	2.0 × 10^2^	[86]
*S.* Typhimurium DT104 and *S.* Typhi	SERS	Assay media	120	100	[87]

## Data Availability

Not applicable.

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
