# Peer review of "Recent Advances in the Detection of Antibiotic and Multi-Drug Resistant Salmonella: An Update"

_ijms, 2021, doi:10.3390/ijms22073499_

Round 1

Reviewer 1 Report

As a review, the paper is well written and the problem is well presented.  The objective of the study and the motivation of the work is also well defined.  This is a very important subject about detection of MDR Salmonella and the emerging technologies.

The paper is accept in present form.

Author Response

Thank for your time

Reviewer 2 Report

The study “Recent advances in the detection of multi-drug resistant Salmonella: An update” provides a revision of emerging technologies that may be used in the detection of Salmonella.

Some general comments (more specific comments are presented in the file in attachment):

The first question regarding this review concerns to the aim itself. Is the aim to review the methodologies for detection of Salmonella, or to evaluate a multidrug resistance profile, or even both? This is not clear along the manuscript and sometimes is really confuse.

Also, the application of the methodologies revised is not clear. Are the methods reviewed for detection of Salmonella in the food chain or in general in diverse samples? When looking at the abstract, the reader may consider that the manuscript revises food related technologies, but then moving on it is not that clear.

Other problem is that the references are inadequate in a large part of the document, it gets better after the flow cytometry section. This inadequacy starts in the introduction, for example, none of the references 1 to 6 refers to what is stated in the text and this follows through the document, where maybe half of the references are not properly used.

Considering that this is a revision of the emergent technologies that may be used for detection of MDR Salmonella, it would be expected a comprehensive revision of the current and emerging technologies, and this does not happen.

The introduction section should be used to introduce the subject in a more direct, objective, and structured form.

Also, several sections are rather complicate to follow, and have several scientific inconsistencies.

Author Response

All the reference misplacements corrected

New references added

Suggestions and expansions applied

New title presented, Abstract revised 

Restructured and new subtitles presented

Update on foods, clinical references, and text removed

Round 2

Reviewer 2 Report

The study “Recent Advances in the Detection of Antibiotic and Multi-Drug 2 Resistant Salmonella: An update” provides a revision of emerging technologies that may be used in the detection of Salmonella.

Despite a considerable improvement of the document, some problems still arise.

Global note: revise the name of species and genes since it should be italicized.

Uniformize the units, we may find CFU/mL, CFU mL-1 or even wrongly CFU/mL-1.

Several references are still inadequate, for example: line 21. Ref 1 and 2; line 41, ref 4-6. All the references should be verified, not only the ones stated by the reviewer.

Figure 1. I am still unable to get to the values of the figure. It seems that the authors summed up all the resistance genes found in the isolates, but one isolate may have more than one resistance gene. In 2014, a total of 262 isolates where analysed so it is not possible to have 564 isolates with resistance genes. Also, the authors should be careful in the statement done since data from 2015 is just from half a year (January to June) and the number of samples is different, maybe it could be referred as a percentage of resistance to each antibiotic. This must be carefully analysed and corrected.

Line 94-98. Please rephrase it, since it seems that the media is differential for different serotypes, and it is not the case.

Line 151. … concluded that the procedure…

Line 162. … antibiotic resistance genes…

Line 218. The acronym MIC is not defined.

Line 267- 273. The paragraph is still confuse, please rewrite it.

Line 347. The paragraph is confuse, please analyse it.

Line 390-392. Please remove this sentence

Line 414-416. Rewrite the sentence, since is not easy to understand what the authors means.

Line 484. … make them…

Line 512. The reference 99 does not refer to Salmonella, further reference 100 is not ok, this is a kit that requires previous growth of the bacterium

Table 1. Revise table 1., the limit of detection can’t be a number inferior to 1. And some of the references (e.g. 70 and 73) are not concerning to detection of resistant serotypes of Salmonella

Verify the list of references, e.g. reference 1 is not complete, reference 2 the title is not complete, and so on.

Author Response

Global note: revise the name of species and genes since it should be italicized. Fixed

Uniformize the units, we may find CFU/mL, CFU mL-1 or even wrongly CFU/mL-1. Fixed

Several references are still inadequate, for example: line 21. Ref 1 and 2; line 41, ref 4-6. All the references should be verified, not only the ones stated by the reviewer. Completed and verified thank you

Figure 1. I am still unable to get to the values of the figure. It seems that the authors summed up all the resistance genes found in the isolates, but one isolate may have more than one resistance gene. In 2014, a total of 262 isolates where analysed so it is not possible to have 564 isolates with resistance genes. Also, the authors should be careful in the statement done since data from 2015 is just from half a year (January to June) and the number of samples is different, maybe it could be referred as a percentage of resistance to each antibiotic. This must be carefully analysed and corrected.

Corrected, re-plotted percentages used as suggested.

Line 94-98. Please rephrase it, since it seems that the media is differential for different serotypes, and it is not the case. Corrected re-organized:

Line 151. … concluded that the procedure… Corrected and reduced.

Line 162. … antibiotic resistance genes… Corrected

Line 218 .... The acronym MIC is not defined. Defined

Line 267- 273. The paragraph is still confuse, please rewrite it. Fixed and reduced.

Lines 390-392. Please remove this sentence. Removed

Line 414-416. Rewrite the sentence, since is not easy to understand what the authors mean? Removed

Line 484... make them... Quotation marks added

Table 1. Revise table 1., the limit of detection can’t be a number inferior to 1. Fixed.

And some of the references (e.g. 70 and 73) are not concerning detection of resistant serotypes of Salmonella. Technically True.

However, we have included S. Typhimurium because the serotype is frequently associated with antibiotic resistance in many fields of food research. 
